# Gene-level metagenomic architectures across diseases yield high-resolution microbiome diagnostic indicators

Braden T. Tierney [1,2,3,4], Yingxuan Tan[1], Aleksandar D. Kostic[2,3,4 ✉] & Chirag J. Patel [1 ✉]

We propose microbiome disease "architectures": linking >1 million microbial features (species, pathways, and genes) to 7 host phenotypes from 13 cohorts using a pipeline designed to identify associations that are robust to analytical model choice. Here, we quantify conservation and heterogeneity in microbiome-disease associations, using gene-level analysis to identify strain-specific, cross-disease, positive and negative associations. We find coronary artery disease, inflammatory bowel diseases, and liver cirrhosis to share gene-level signatures ascribed to the *Streptococcus* genus. Type 2 diabetes, by comparison, has a distinct metagenomic signature not linked to any one specific species or genus. We additionally find that at the species-level, the prior-reported connection between *Solobacterium moorei* and colorectal cancer is not consistently identified across models—however, our gene-level analysis unveils a group of robust, strain-specific gene associations. Finally, we validate our findings regarding colorectal cancer and inflammatory bowel diseases in independent cohorts and identify that features inversely associated with disease tend to be less reproducible than features enriched in disease. Overall, our work is not only a step towards gene-based, cross-disease microbiome diagnostic indicators, but it also illuminates the nuances of the genetic architecture of the human microbiome, including tension between gene- and species-level associations.

[1] Department of Biomedical Informatics, Harvard Medical School, Boston, MA, USA. [2] Section on Pathophysiology and Molecular Pharmacology, Joslin Diabetes Center, Boston, MA, USA. [3] Section on Islet Cell and Regenerative Biology, Joslin Diabetes Center, Boston, MA, USA. [4] Department of Microbiology, Harvard Medical School, Boston, MA, USA. ✉email: Aleksandar.Kostic@joslin.harvard.edu; chirag_patel@hms.harvard.edu

The ecology of the human microbiome is known to be associated with both phenotype and environment[1,2]. Here, we introduce "microbiome architectures", which, analogous to human genetic architectures[3], are the characteristics of the microbiome, which, as a group, correlate with human phenotype. More specifically, we compute architecture by identifying the complete set of associations between the microbiome and a given host disease. We hypothesize that these could potentially be jointly diagnostic for different aspects of host health[4-7]. Moreover, identifying common—and distinct—architectures across diseases can shine light on the degree to which diseases share common etiologies. Achieving these ends, however, requires identifying how architecture changes across an array of human diseases in a manner that can easily be tested with in vivo or in vitro experiments.

Others have considered microbial community ecology across human individuals. Outside of single-disease metagenome-association studies, investigators have introduced the concept of the "enterotypes," 3 hypothetically phylogenetically (at the phylum- and genus level) and functionally distinct groups of microbiome compositions. Enterotypes were initially identified across individuals from different backgrounds and countries[8,9]. While inter-individual microbiome variation and the presence of enterotypes is debated, their contribution to the field of comparative meta-genomics remains fundamental[10-12]. However, by comparing microbiome ecology across a range of host phenotypes, the concept and construction of architectures sidesteps the challenge of building grand views of a "normal" microbiome. Architectures instead enable the identification of specific (but still holistic) microbial factors associated with specific host phenotypes across sources of metagenomic variation.

At the heart of a metagenomic architecture rests a set of statistical associations between individual microbial features (e.g., species, pathways, or genes) and a given human phenotype. These associations are subject to the same challenges of any observational study, such as lower sample size (lack of power to detect associations) or confounding (e.g., not accounting for particular batch effects, geography, and/or diet). Lack of power and bias in observational studies (such as confounding) can lead to type 1 (false-positive) and type 2 (false-negative) errors.

Many studies use "meta-analyses" to aggregate and compare results across cohorts. There are a few approaches for carrying out a meta-analysis (e.g., random vs. fixed-effect meta-analyses[13]), and they provide a way to estimate an "overall" association size. Historically, they have been deployed for both randomized and observational research[14], such as to aggregate effects across clinical trials[15]. Meta-analyses are emerging in the microbiome and have been used to discover new microbiome-disease associations in, for example, colorectal cancer[6,16-18].

However, meta-analyses are still potentially subject to confounding effects due to variable model specification strategies that occur in individual studies. In most situations, investigators choose a set of measured and potential confounding variables to adjust for in a model based on a prior hypothesis between the nature of the association between the independent and dependent variables. However, when the exact mechanism of potential confounding is unknown, the choice of potential measured confounders to adjust for in a model is arbitrary. The degree to which variation in model specification (e.g., adjusting for certain confounders and not others) changes the relationship between dependent and independent variables has been described as "Vibration of Effects" (VoE)[19-21]. Modeling VoE enables researchers to identify not just that a result is irreproducible among certain model specifications, but potentially why that is the case[20]. Briefly, we hypothesize that the larger the variation of associations that occur due to measured confounding and model choice, the less robust an association is. In other words, a robust association should persist across all or most configurations of study designs and model choices.

To be clear, we do not claim to identify the "best" method for computing architectures. Rather, we aim to propose architectures as a concept and demonstrate one method for their identification that controls for inconsistency in model output due to model specification. There are many options for computing the association between a disease and microbiome feature, ensuring these associations are robust, and meta-analyzing across datasets. Each of these steps rests upon volumes of biostatistics and microbiome literature. For example, a range of methods are used in the microbiome, from nonparametric tests to complex machine learning, like random forests.

Here, we propose one avenue for the identification of robust, multidata-type, microbial architectures in human disease by applying an analytic framework that considers a vast array of model specifications. We quantified the shared and distinct microbiome-disease architectures for seven prevalent diseases. We used the results of our meta-analysis and VoE pipeline to build high-resolution, robust multidisease architectures for seven diseases (adenoma, colorectal cancer (CRC), liver cirrhosis (CIRR), inflammatory bowel diseases (IBD), type 2 diabetes (T2D), otitis, and atherosclerotic cardiovascular disease (ACVD)), with a novel emphasis on gene-level, cross-disease associations. We specifically chose to examine otitis as a form of negative biological control, as, to our knowledge, it has limited reported association with the gut microbiome, and we expected it to have a negligible metagenomic architecture.

## Results

**Microbiome meta-analyses alone yield a panoply of associations.** We executed a combined meta-analytic and VoE approach to identify microbiome-based associations with seven common diseases spanning 13 cohorts and 2573 samples (Fig. 1 and Supplementary Data 1). We identified associations between microbial taxonomy/pathway/gene family abundance and disease presence, defining statistically significant relationships as those with a false-discovery rate (FDR)-adjusted $P$ value of <0.05. A glossary of terms related to our pipeline and regression analysis is listed in Supplementary Data 2.

The results of our initial analysis demonstrated how associations vary as a function of data type, disease, and phenotype (Figs. 1A and 2). We used a linear modeling approach to identify initial associations between each microbial species ($N = 6832$), pathway ($N = 76,251$), and gene family ($N = 1,167,504$), fitting "maximal models" containing the metadata variables present for a given phenotype within a given dataset ("Methods", Supplementary Data 1). We meta-analyzed the within-cohort regression results across the three phenotypes (CRC, adenoma, and T2D) that were represented across multiple independent cohorts. Across all phenotypes, gene families had, on average, the most FDR-significant associations (mean = 16,557 (1.4% of all tested)), followed by pathways (mean = 279 (0.4% of all tested)), followed by species associations (mean = 31 (0.05%)). Meta-analyzed phenotypes had, on average, fewer FDR-significant associations (mean = 2890) than non-meta-analyzed phenotypes (mean = 6673).

Otitis and adenoma had the fewest number of FDR-significant associations (49 and 90, respectively). Conversely, we found IBD, ACVD, CRC, and T2D to have the largest number of FDR-significant associations (62,563, 27,573, 21,197, and 1258 respectively). Of all statistically significant associations, 107,966 were positively associated with phenotype (i.e., increase in abundance in the presence of disease) and 9762 were negatively associated.

**Vibration-of-effects' stress-test microbiome associations.** We next sought to test the robustness of our identified associations

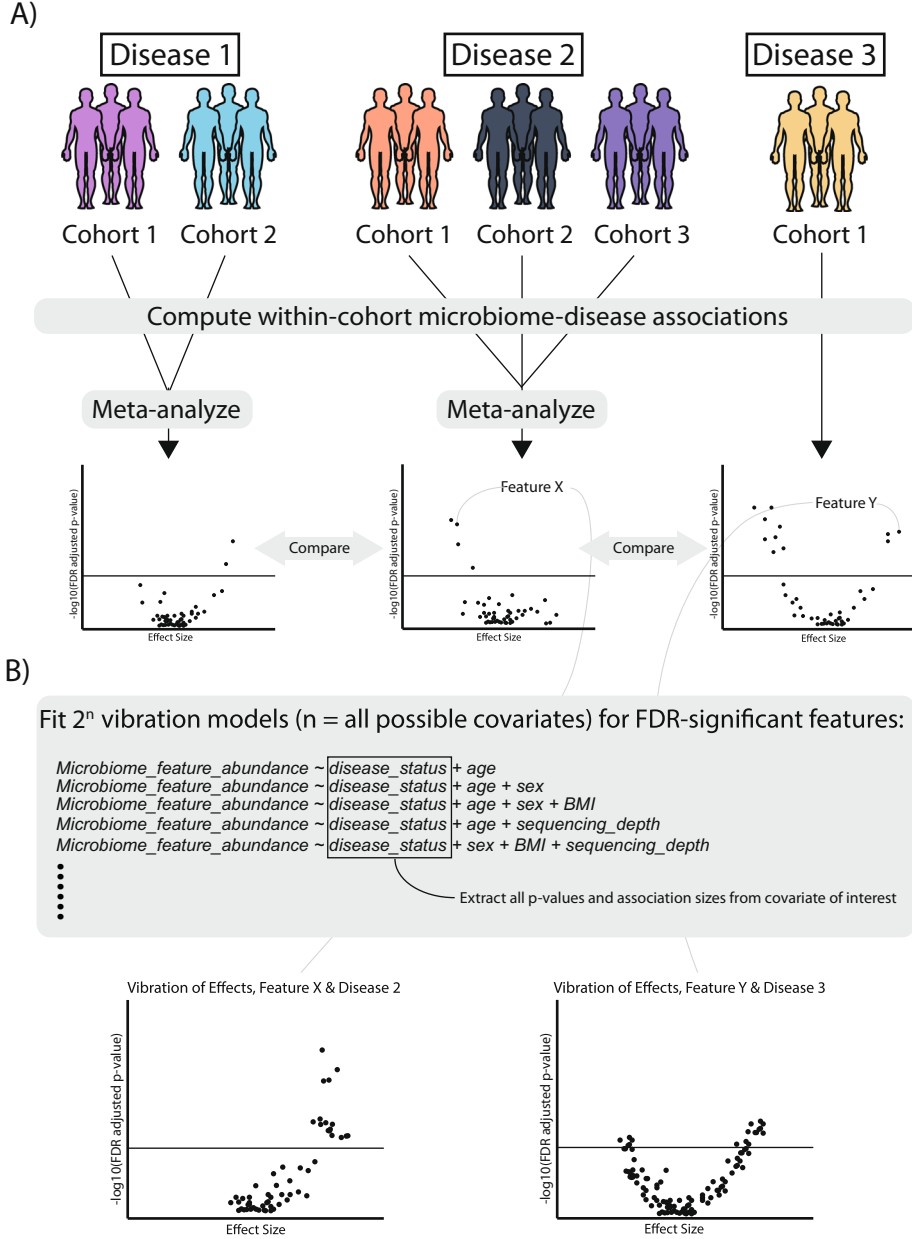

**Fig. 1 Pipeline overview. A** Using publicly available metagenomic, shotgun sequencing datasets, we computed linear associations between microbiome features (gene family, pathway, or species abundances) and for each of seven different diseases separately. In cases where multiple cohorts were present for one disease, we meta-analyzed the association output. **B** We then computed Vibration of Effects (VoE), where, given the available individual-level metadata, we determined how model specification changes the association between each false-discovery rate significant feature and host phenotype.

via modeling VoE. For each FDR-significant feature, we fit every possible model specification given the possible potentially confounding variables measured in the individual cohort and assessed how association sizes and *p*-values change as a result of model specification choice. For each dataset, we fit every combination of covariates found in the maximal model ("Methods", Fig. 1B), quantifying the ranges (between the 99th and 1st percentile) in estimate and *p*-value size for the independent, binary disease status variable. We additionally quantified Janus Effects (JEs), which we define as the fraction of estimates with positive associations for a given feature (as opposed to negative). For example, a JE of 0.5 is an example of a "non-robust" association. For a given microbial feature with a JE of 0.5, half of its associations with disease were negative and half were positive. In total, we tested VoE for 117,966 features that were reported in the literature as disease-associated and/or found to be FDR-significant in our initial step, fitting a total of 67,600,562 models. We found the mean estimate ranges and JEs to be $0.36 + -0.31$ and $0.92 + -0.27$ (Supplementary Fig. 1), respectively.

We then selected for "robust" disease–microbiome associations (Fig. 3). We filtered for associations with minimal vibration of effects, with JEs of >0.99 or <0.01. Overall, this filtering step removed 1717 (1688 gene families, 9 pathways, and 4 species) of the initial FDR-significant associations (1.4%) across all diseases. ACVD, cirrhosis, and otitis did not present any associations with JEs that failed this initial filtering criterion. We additionally tested more stringent filters, taking associations not only with low JEs, but also those in the bottom 75%, 50%, and 25% quantiles of estimate ranges (i.e., selecting for associations with consistent estimate sizes across all models), yielding further reductions to

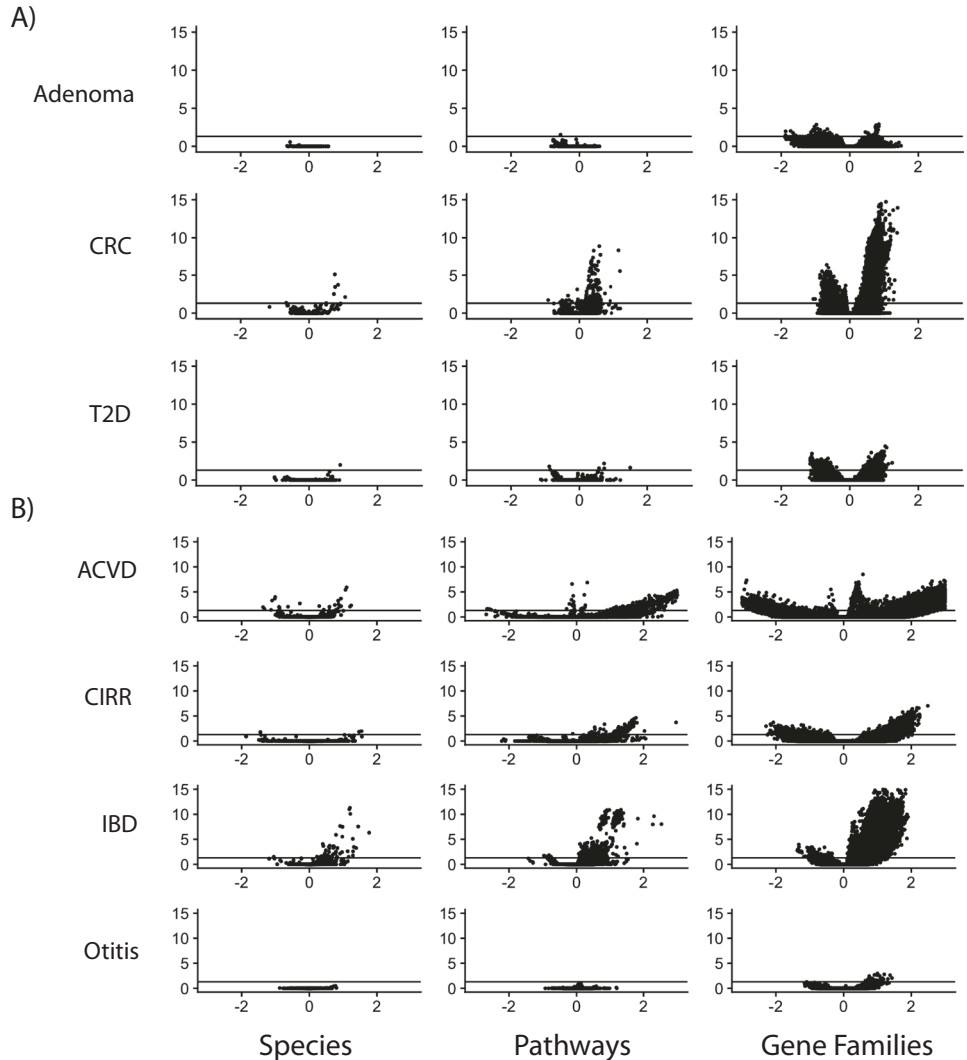

**Fig. 2 Initial association output.** Initial association outputs for each **A** meta-analyzed and **B** single-cohort phenotype, split by species associations, pathway associations, and gene family associations. Each point represents a different feature (e.g., species). $Y$ axes are false-discovery rate-adjusted $\log_{10} P$ values. Solid line is false-discovery rate-adjusted statistical significance ($P < 0.05$). $X$ axes are the beta-coefficient on the binary, independent disease variable of interest.

87,012 (73.8% of all initially FDR-significant features), 43,506 (37.0%), and 10,877 (9.2%) features remaining, respectively (Supplementary Data 3).

We compared our meta-analysis and VoE output to a range of alternative modeling strategies: univariate associations, associations with a fixed effect adjusting for cohort (for the multicohort phenotypes), elastic net regression, random forest regression, and sparse partial least squares (sPLS) regression (Supplementary Fig. 3). We additionally randomized disease status for all diseases as a negative control. We found that architectures (a) built on randomized disease indicators (for cases and controls) did not overlap with architectures we identified on non-randomized data (Supplementary Fig. 4), and the (b) other approaches yielded similar importance in variable ranking across all data modalities and phenotypes. Our results indicated that generally the same features are being implicated in diverse modeling approaches and that modeling VoE is the most conservative option (Supplementary Fig. 5). Batch-adjusted regressions yielded the most similar results in terms of statistical significance to our approach, followed by univariate models. For the comparison to variable selection methods, sPLS and elastic nets were the most similar to

our method. Overall, we found that the ranking of features—especially among those flagged as significant and robust by our pipeline (mean Spearman correlation with our initial ranking of features $= 0.88 +/- 0.01$)—was similar across all methods (Supplementary Fig. 6).

**Gut microbiome-disease architectures are dependent on data type.** Having identified a set of robust associations between seven human diseases and microbial gene families, pathways, and species, we next sought to use our results to comparatively investigate gut microbiome-disease architectures as a function of data modality. We compared architectures between diseases by estimating the degree of similarity of associations between phenotypes. We found that of the 6344 microbial species, pathways, or genes that were associated with multiple phenotypes, 798 (12.5%) had association sizes with opposite signs. After filtering these out and comparing the overlap between robust and statistically significant features across all diseases for all combined data modalities, we found moderate overall shared microbiome architecture (Supplementary Fig. 3). The greatest overlap of statistically significant features was between ACVD and IBD ($N =$

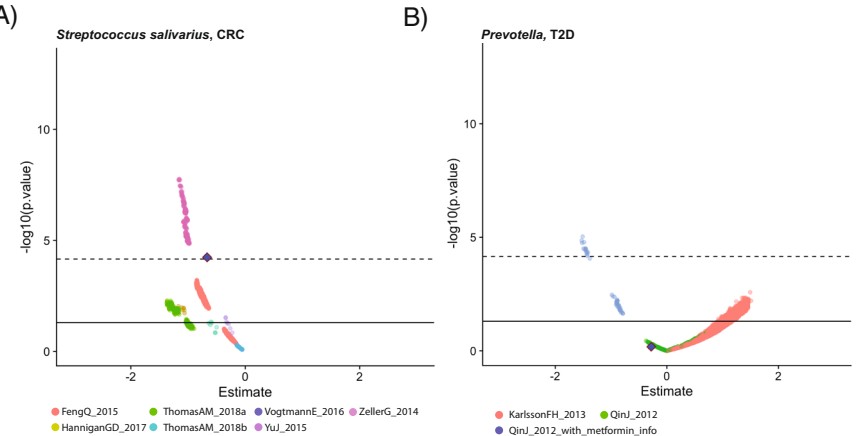

**Fig. 3 Examples of associations of varying strength.** Example of a robust (**A**) and nonrobust (**B**) association as identified by modeling vibration of effects. Each point represents the association deriving from multiple linear regression between the disease and microbial feature of interest for a different modeling strategy. $Y$ axes are nominal $\log_{10} P$ values. Solid line is nominal statistical significance ($P < 0.05$). $X$ axes are the beta-coefficient on the binary, independent disease variable of interest. Dotted lines represented the false-discovery rate-adjusted $P$ values. Point colors correspond to cohorts. The solid blue diamond marks the $P$ value and estimates achieved through meta-analysis across all cohorts. The species listed in **A** was included in the downstream analysis, as it exhibited meta-analytic false-discovery rate (FDR)-adjusted statistical significance, whereas the species in **B** was not included, as it was not FDR-significant and was not robust, as nominally significant opposite sign results (a Janus Effect) could be achieved with differing model specifications.

1996), followed by ACVD and CIRR ($N = 1290$), and CRC and ACVD ($N = 984$). In total, 147 features were shared between exactly 3 phenotypes, and 5158 features were shared between exactly 2. In total, two features were shared between four phenotypes (IBD, ACVD, CRC, and CIRR).

We identified what has been described as the "Anna Karinena effect"[22,23]—that unhealthy microbiomes are not alike—to be generally true at the species and pathway level but not entirely at the gene level (Fig. 4). For example, at the species level, ACVD, CIRR, IBD, and CRC clustered together, whereas T2D was entirely distinct. At the gene family level, however, not only did the ranking of similarity change between these four diseases, but T2D was also grouped in with them. In total, nine taxonomic (MetaPhlAn2) annotations were associated with ACVD and IBD (members of the genera *Solobacterium* as well as *S. Moorei*, the families *Lactobacilliaceae* and *Actinomycetaceae*, the genus *Erysipelotrichaceae*, and members of the genus *Streptococcus*), one with IBD and CRC (the genus *Peptostreptococcus*), and another, *S. anginosus*, with all IBD, CIRR, and ACVD.

In total, 95 pathways were conserved between diseases. We found only one pathway, phosphopantothenate biosynthesis I, to be negatively associated with multiple diseases: ACVD and CIRR. The species annotated to the disease-associated pathways were similar to those in the species-level results above. IBD and ACVD overlapped in 50 pathway annotations, the majority of which mapped to various *Streptococcus* species. The 8 pathways conserved in association with CRC, ACVD, and IBD all mapped to *Solobacterium moorei*.

**Gene-level taxonomic analysis identifies broad-spectrum health and disease-associated strains.** In total, 5204 (82.0%) of the microbial features associated with multiple diseases were gene families. In total, 3728 genes had taxonomic annotations in the UniProt database, which we mapped to 221 distinct species. We visualized these annotations in a taxonomic tree to better understand the shared phylogenetic trends within our diseases of interest (Fig. 5). We hypothesized that while individual gene annotations could be spurious (due to the challenge assigning taxonomies to single-gene families), the overall trend toward different phylogenies across all genes would inform disease-

architecture structure. In total, 181 species (82.0%) and 3504 genes (94.0% of those with annotations) mapped to the *Firmicutes* phyla. Of these, the strain *Solobacterium moorei F0204* (662 genes (18.1%), associated with IBD (6/662), ACVD (658/662), and CRC (662/662)), genera *Streptococcus* (36 species, 553 genes (14.2%), associated with IBD (380/553), ACVD (527/553), CRC (10/553), and CIRR (243/553)), and *Clostridium* (23 species, 233 genes (6.3%), T2D (84/233), IBD (143/233), ACVD (65/233), CIRR (1/233), and CRC (175/233)) dominated. All of the genes reported here and in the proceeding sections, including their names and UniProt annotations, can be located in Supplementary Data 4.

We have discovered enhanced prior-reported associations with disease by integrating gene- to strain-level information. We identified gene clusters that mapped specific species genomes or pan-genomes. Over 100 genes associated with multiple diseases were annotated as mapping to *S. moorei*. Other annotations with large numbers of genes mapping to them were *Erysipelotrichaceae bacterium CAG:64* (CIRR (1/810 genes), ACVD (802/810 genes), IBD (806/810 genes), and T2D (19/810 genes)), *Firmicutes bacterium CAG:102* (associated with IBD (517/517 genes) and CRC (517/517 genes)), and *Gemella haemolysans* (associated with CRC (108/110 genes), ACVD (110/110) genes, and IBD (2/110) genes). While certain studies have reported these organisms as associated with these diseases (specifically IBD, T2D, and CRC), their shared, strain-specific markers have not until now been identified[6,17].

Moreover, we found our gene-level associations to be, in some cases, strikingly more robust than species-level associations, such as in the case of the association between *S. moorei* and CRC. *S. moorei* comprises at least four strains—an association at the species-level alone yields a Janus effect (Fig. 6). All the genes in Figure 5 from *S. moorei* that were associated with CRC, however, had the same strain-level annotation (*S. moorei F0204*), and none of them exhibited a Janus effect.

We also identified select clusters of genes that appeared to be negatively associated against multiple diseases. These did not fall within any single phyla or otherwise coherent taxonomic grouping. The majority of the groups contained fewer than ten genes, with the exceptions being *Eubacterium sp. CAG:86*

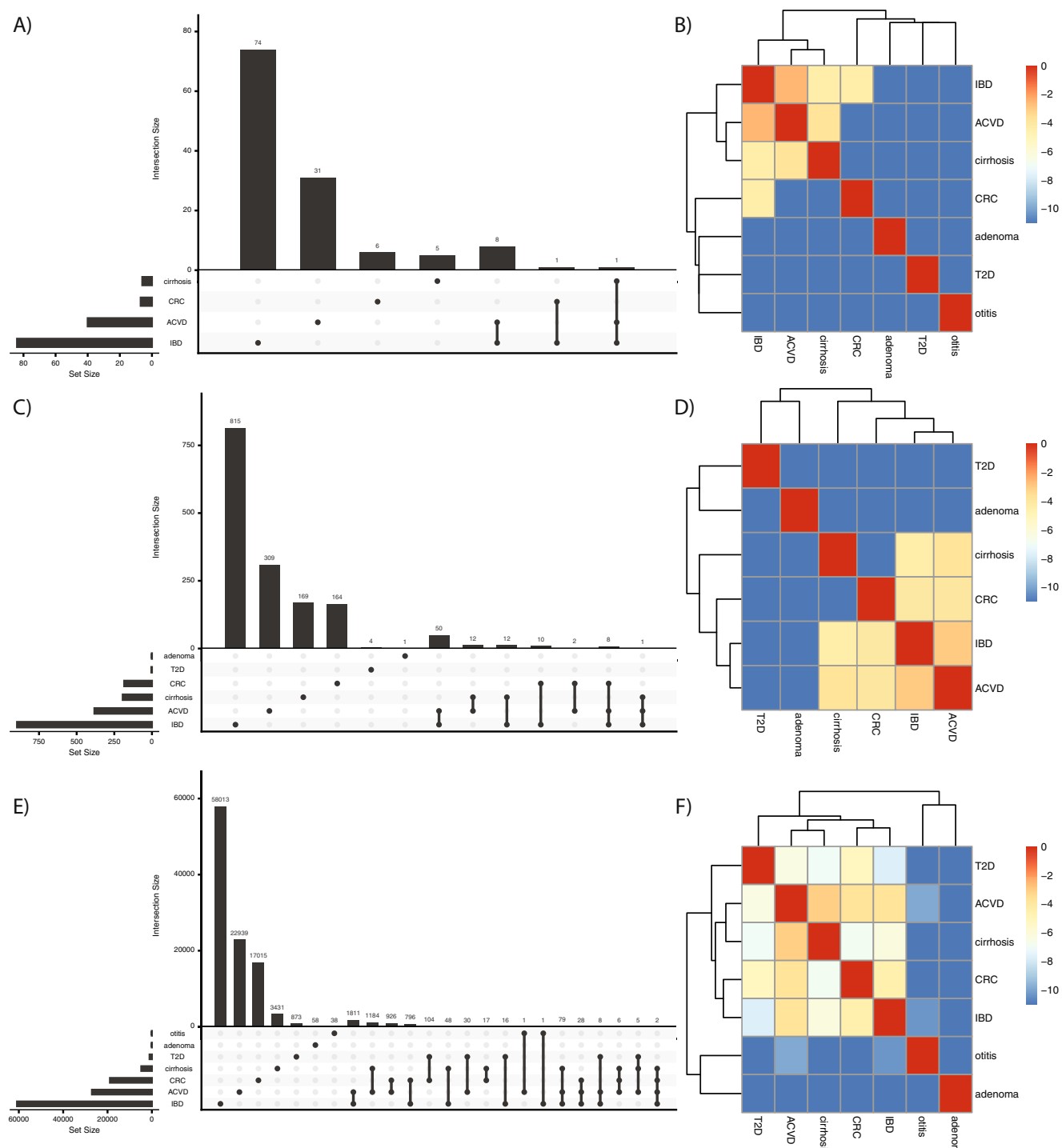

**Fig. 4 The disease architecture of the human microbiome across seven phenotypes as a function of data modality. A**, **C**, **E** describe the species, pathways, and gene families associated with each phenotype and the overlap therein, respectively. **B**, **D**, **F** show the natural log of pairwise jaccard similarity between binary vectors indicating all of the features (e.g., species or pathways or gene families) associated with a given phenotype.

(negatively associated with T2D (10/55 genes), ACVD (54/55 genes), CRC (1/55 genes), and CIRR (50/55 genes)), *Coprobacillus sp. CAG:235* (negatively associated with T2D (10/10 genes) and CRC (10/10 genes)), and *Sutterella sp. CAG:351* (negatively associated with ACVD (22/22 genes) and CRC (22/22 genes)). We were unable to find a specific functional trend among these groups of health-associated genes and their particular function in microbiome–host interactions warrants further investigation (Supplementary Data 4).

Overall, visualizing gene-level annotations on a taxonomic tree further informed the taxonomic delineations between disease. For example, 543 out of 1920 (36.6%) gene overlaps between CIRR, ACVD, or IBD associations were in the *Streptococcus* genus. T2D was not associated with a high-resolution taxonomic group, but rather all of its shared associations fell within the orders *Bacteroidales* (1 species, 3 genes), *Clostridiales* (11 species, 116 genes), and *Erysipelotrichales* (4 species, 31 genes).

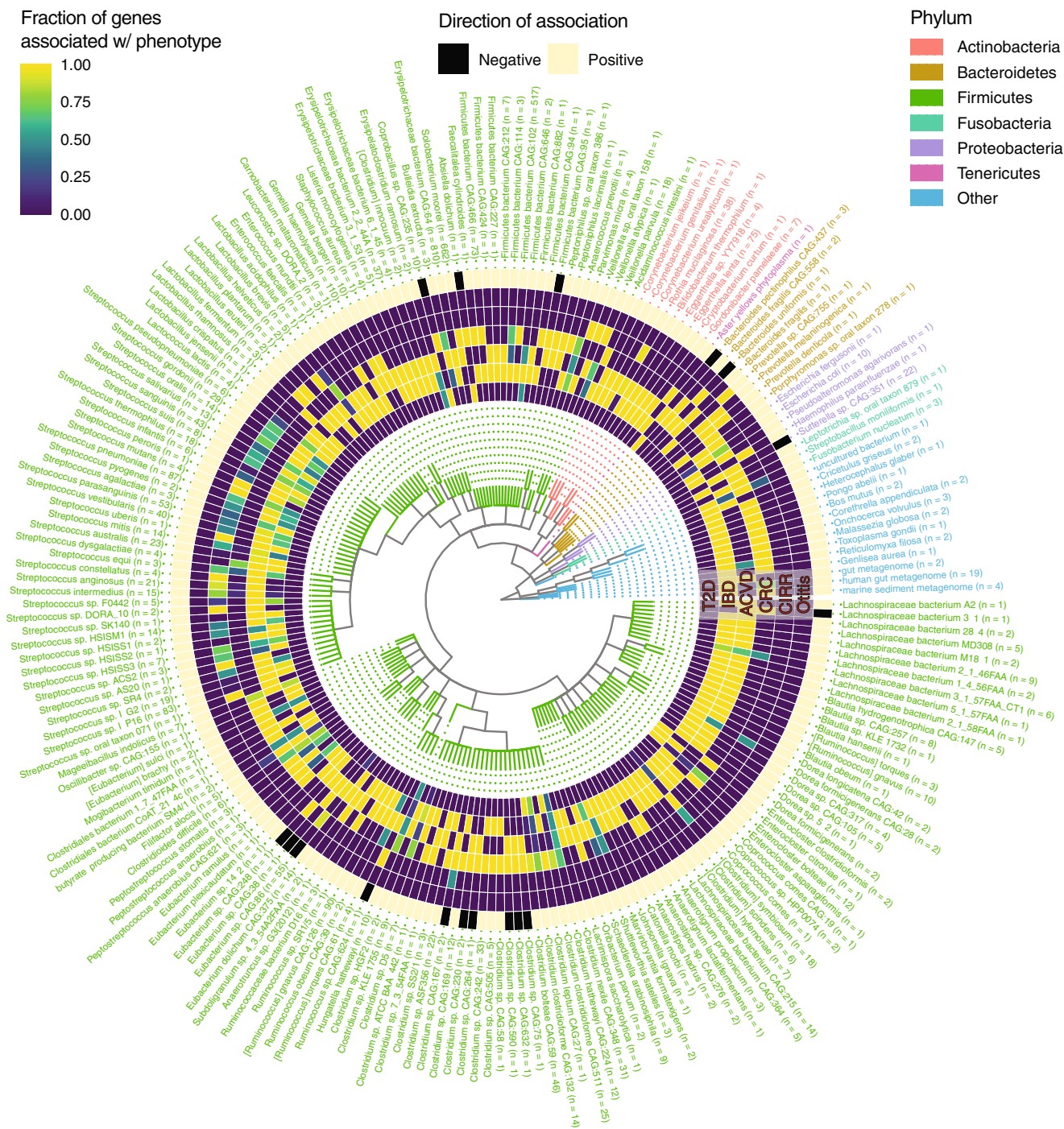

**Fig. 5 Cross-disease, gene-level architectures.** The taxonomic distribution of genes associated with at least two phenotypes, excluding adenoma due to a lack of significant, robust, and overlapping associations. Heatmap color for the inner six rings corresponds, for a particular phenotype, the fraction (of the N in the axis labels) of genes associated with a given taxonomic annotation. Lighter colors indicate a value closer to 1. Outer ring corresponds to if an association was negative (black) or positive (beige) across all phenotypes. Text color corresponds to phylum, with nonbacterial phyla listed as "Other".

**Gene-level metagenomic architectures are reproducible across additional cohorts.** We next aimed to reproduce our gene-level architectures for CRC and IBD in independent cohorts[18,24]. For IBD, we analyzed 1338 samples from 106 subjects (80 cases, 26 controls), and for CRC, we analyzed 82 samples from the same number of subjects (22 cases, 80 controls). For the 61,575 genes that had representative sequences in UniRef, were FDR-significant, were robust, and were associated with IBD, 2104 (3.4%) were nominally significant in the validation cohort. For the 19,498 genes that were FDR-significant, robust, and associated

with CRC, 4688 (24.04%) were nominally significant in the validation cohort (Fig. 7A). Of all the associations tested, 10,567 (54.20%) and 22,616 (44.0%) in the validation cohorts had the same direction of association as those in the initial CRC and IBD cohorts. Positive associations with disease were more likely to reproduce than negative in both cases (Fig. 7B, C).

We examined the taxonomic annotations for the genes, counting the number of annotations to each species/strain, and identifying the top 25 for the initial and validation cohorts (Fig. 7D, E). We found that the top annotations were similar

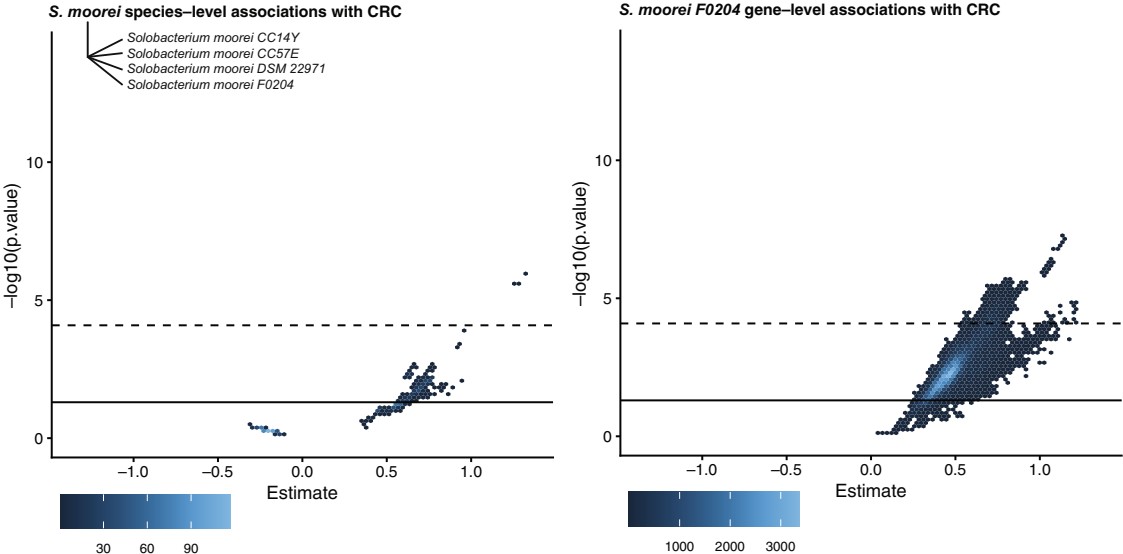

**Fig. 6 Vibration of an association with CRC at the species versus gene level.** Each point corresponds to a different linear model specification. *Y* axes are $\log_{10} P$ values. Solid line is nominal statistical significance ($P < 0.05$). *X* axes are the beta-coefficient on the binary, independent disease variable of interest. The dotted lines correspond to the 0.05 false-discovery rate-adjusted $\log_{10}$ cutoff (for species and genes, respectively) in our original meta-analysis, whereas the solid line corresponds to the nominal ($P = 0.05$) cutoff. The gene-level plot (right) contains the overlaid vibration output for every *S. moorei* gene ($N = 662$, all deriving from one strain) that was significantly associated with CRC.

between the two, with 22 of the top 25 annotations in the validation cohort being present in the initial cohort for CRC. In total, 16 of the top 25 annotations in the validation cohort were present in the initial cohort for IBD. Overall, this indicates that the gene-level architectures, when mapped to strains/species, were reproducible in new cohorts.

## Discussion

Anchoring architectures relative to host disease (as opposed to enterotypes, which are not linked to disease) enables a biomedicine-focused view of global human microbiome structure. We used the output of our meta-analytic and VoE analysis to quantify the degree to which disease-associated microbiome architectures are similar or dissimilar, identifying shared, gene-cluster-specific signatures of disease. Overall, we found striking and previously unrecognized high-resolution genetic and taxonomic signatures associated with ACVD, IBD, CRC, and cirrhosis, reproducing our results for CRC and IBD. We additionally found limited associations with otitis, which we hypothesized would be the case due (to our knowledge) to its limited reported biological link to the gut microbiome.

The hypothesized dissimilarity between diseased microbiomes has been dubbed the "Anna Karenina Effect," with all analysis to date being carried out at the species level[16–18]. Our work extends these past efforts, showing that gene-level analysis reveals microbiome signals that show greater similarity overall in microbiome-disease structure than species-level analysis. We generally reproduce some prior findings—like the presence of *Firmicutes*, specifically *Clostridiales*, that are broadly disease-associated[18]. We were able, however, to take this work a step further, identifying pan-disease-associated gene clusters.

Our gene-level architecture analysis captured a previously undocumented strain-level exploration of pan-disease-associated microbes. For example, consider *S. moorei*, reported to be associated with CRC and ACVD[6,25]. We found the reference-based species-level annotation for this organism to exhibit a Janus effect (i.e., was not robust to model specification and study design choices). We did, however, identify a cluster of 622 *S. moorei*

genes as indicative of not only ACVD and CRC, but also IBD, all deriving from a single strain, F0204 (one of 4 strains listed in NCBI's taxonomic database)[26]. This may indicate that certain strains of *S. moorei* are disease-associated, whereas others are not. By identifying specific genes, we provide specific markers of these strains.

In addition, the high-resolution to low-resolution mapping that occurs when connecting gene-level analysis to taxonomies provided glimpses at the overall, strain-specific picture of metagenomic architectures that may be worth following up in a laboratory setting. Examples include the multi-disease-associated *Gemella haemolysans*, *S. moorei*, *Erysipelotrichaceae*, and *Streptococcus*, as well as the potentially broad-spectrum health-associated *Eubacterium sp. CAG:86*, and *Coprobacillus sp. CAG:235*, *Sutterella sp. CAG:351*.

Focusing on the gene level may have an additional practical advantage over analysis at the species- or pathway level in the clinic: it allows for high-throughput, multiplexed, PCR-based, and specific diagnostics. Leveraging conserved (or distinct) disease-associated gene families could potentially yield multifactor diagnostics, where a single assay could screen disease risk for multiple phenotypes simultaneously. In contrast, a species-level diagnostic, we hypothesize, across multiple taxonomies will not be disease-specific nor necessarily easy to design a test for.

Methodologically, combining a meta-analysis with modeling vibration of effects enables the identification of disease architecture robust to model specification and study design. Meta-analyses in microbiome research—compiling and comparing associations across cohorts, usually at the level of species abundances—have proven a powerful approach to increasing the confidence of host–microbiome associations, notably in the case of colorectal cancer (CRC)[16,17,27,28]. However, while powerful, they are not bias-proof. They are still subject to many of the same pitfalls that render single-cohort associations irreproducible, such as failing to adjust for measured confounding. Therefore, as a secondary result, our efforts yielded an "association-prioritization" framework to assist biologists who attempt to functionally validate the associations that emerge from observational studies in costly in vivo experiments. The associations generated via

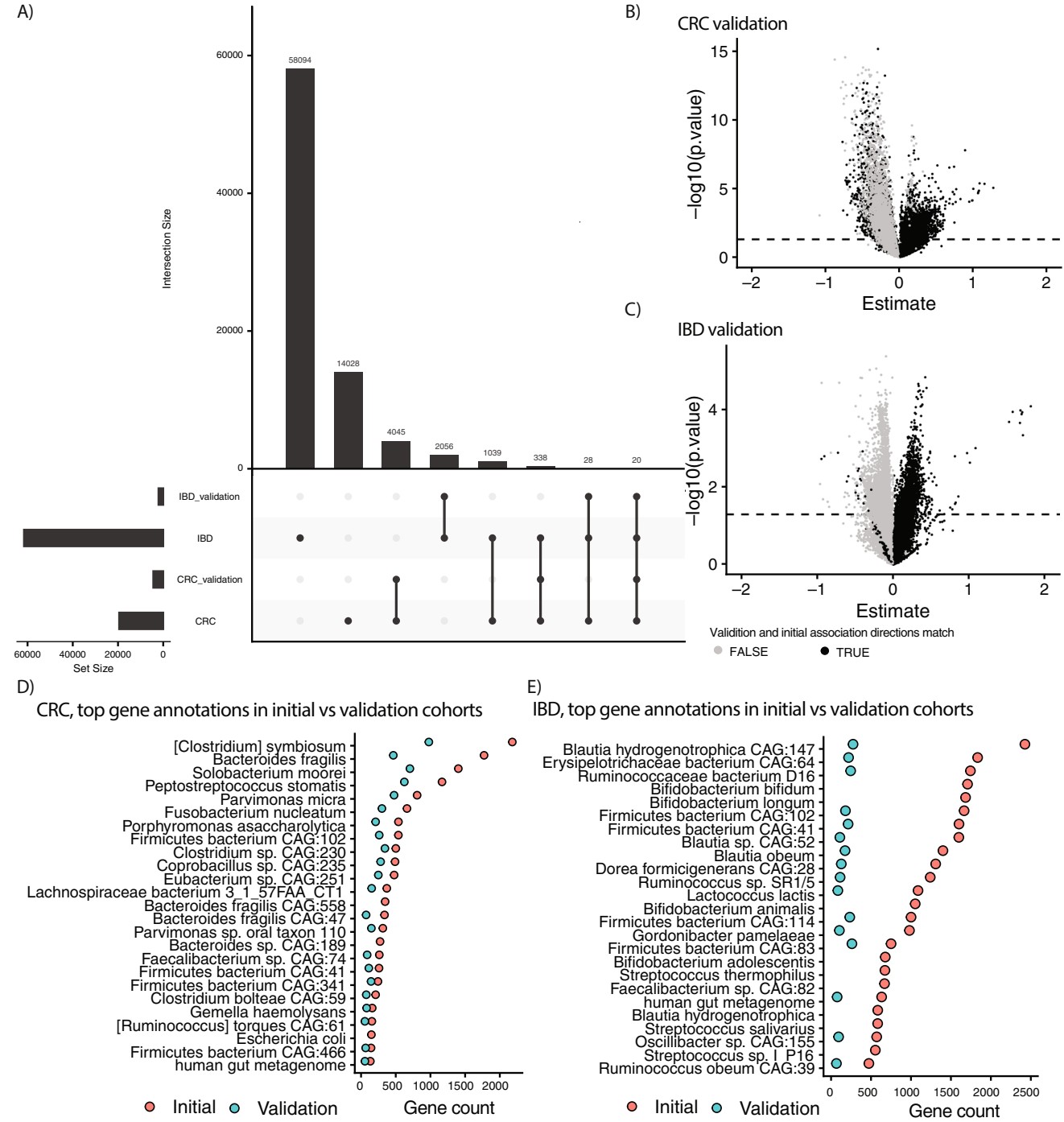

**Fig. 7 Validation of gene-level architectures for CRC and IBD.** We used unadjusted, univariate linear models to test the association between gene abundance and disease state for the genes associated with IBD and CRC in two cohorts not analyzed in our initial study. **A** Overlaps between the initial and validation cohorts in terms of significant genes. **B**, **C** Volcano plots of CRC and IBD estimate sizes and nominal log$_{10}$ P values for validation cohorts. Each point represents a different gene family. Dotted line is nominal (*P* value < 0.05) significance. Exact *P* values < 0.05 are shown above the line. *Y* axes are nominal log$_{10}$ *P* values. *X* axes are the beta-coefficient on the binary, independent disease variable of interest. Black dots indicate the association direction (e.g., positive vs. negative) matched in the initial cohort(s), gray indicates initial, and validation associations did not have the same direction. **D**, **E** The top 25 taxonomic annotations (by frequency) of genes associated with CRC and IBD in the initial cohorts and how many of these are also found in the top 25 annotations of genes in the validation cohorts.

multicohort meta-analysis, filtered via VoE, and reproduced in an external cohort ought to be prioritized first, followed by associations that have only gone through a subset of these stringent checks (e.g., only meta-analyzed data, followed by single-cohort data). Conversely, we would assign low priority to associations that are difficult to interpret.

Of course, our approach is not without drawbacks. First, it is extremely conservative: we may be filtering out true-positive results, particularly at the species- and pathway level. For example, it relies on exclusively linear modeling due to its speed and the ease of performing statistical inference on the associations that are output. Many microbiome studies rely on random

forests, which can capture nonlinearities but can be difficult to perform inference on (e.g., ascertain the error around the association estimate) or interpret (e.g., propose a direction of association)[29–31]. It is possible that a more sophisticated modeling approach would be able to find more associations. We did demonstrate that alternative modeling approaches—like the elastic net, random forest, and sparse PLS, tend to yield similar results for the features our approach identified. Moreover, we made choices in our modeling strategy—like averaging longitudinal samples from the same individual—due to issues of scaling mixed-effect models to the needs of our study—that could affect statistical power. Finally, an additional drawback of the data underlying our findings is its reliance on compositional data (i.e., relative abundances). Many have raised issues regarding potential bias of batch effects of relative abundance data (which is a confounder like other variables assessed in our VoE approach). As our ability to ascertain true microbial population size improves via sequencing technology and statistical methods, we should consider leveraging alternative tools or data structures in building architectures to reduce this source of bias[32,33].

For these reasons, we do not claim that our method is the only one to identify architectures—rather, the value of our work lies in demonstrating the utility of (specifically, gene-level) architectures for comparing and reproducing disease signatures across the microbiome. As with any statistical challenge facing a plethora of analytic choices, the "best" method for identifying those genes remains to be determined and must be the focus of future work.

Overall, this work depicts a path for researchers for moving microbiome associations from the abstract to the robust. In short, fitting and reporting a single model is simply not sufficient. However, if we are able to identify robust-to-specification associations that reproduce across cohorts, we will increase the efficiency of biomedical experiments.

## Methods

**Data collection**. We accessed all the data used in this project from curatedMetagenomicDataV1.14.1[34] and its associated R package. curatedMetagenomicData contains HUMAnN2[35] V0.7.1 output for each sample in 69 shotgun sequencing cohorts spanning 5 body sites. We downloaded species-level abundances, pathway-level abundances, and gene family-level abundances for each dataset, generating single dataframes for each microbiome data type. Overall, we aggregated 57 cohorts, 10,199 samples, and information on 45 different disease states. We chose to only analyze diseases in patients who had samples from the gut microbiome that had either (1) multiple cohorts containing cases and controls or (2) >100 cases. We chose to exclude one disease, infectious gastroenteritis, from our list, as despite having multiple cohorts, as it only had data from fewer than 20 individuals and had limited associations reported in the literature. We included the Type 1 Diabetes data from curatedMetagenomicData in our initial regressions; however, family-based structure of one cohort (i.e., one cohort contained data on related individuals) and low sample size of cases prompted removal, specifically to ensure proper harmonization across case–control studies. As such, we were left with a total of 2573 samples, 13 cohorts, and 7 diseases. We collected additional sample information on metformin usage from the Qin study[36] from Forslund et al.[37] This split the Qin study[36] into two sample subsets, one subset where the samples had metformin information and another that did not have metformin information.

**Filtering gene family data**. Upward of 4 million gene families were identified by HUMAnN2 across all of the datasets in curatedMetagenomicData. We filtered these data given that our pipeline requires computing an individual association for each gene, and that highly rare genes were unlikely to generate statistically meaningful results in our modeling process. We only selected genes that were (1) present in at least 2 of the 57 cohorts in curatedMetagenomicData and (2) present in at least 10% of all 10,199 samples. After applying these filters, we were left with 1,167,504 gene families.

**Computing diversity and richness metrics**. We computed Shannon diversity using R's Vegan package[38] V2.5.6 for each individual and each microbiome data type (species, pathways, and gene families). That is to say, "gene family diversity" is Shannon diversity computed with gene family abundances. We computed genus richness for each sample by counting the number of genera with nonzero abundance in the species dataframe.

**Variable selection for maximal models**. We first selected the covariates to include in our initial, maximal model and eventually vibrate over. For each cohort, we selected covariates that recorded data for at least 90% of samples. We also removed variables that were singular or co-occurring with disease presence.

**Computing initial microbial feature associations**. We fit a linear model using base-R's lm function within a cohort for each microbial feature (species, pathway, and gene family) using the covariates identified in our maximal models (Eq. (1)).

$$ln(microbial\_feature_i + f) \sim disease\_state + covariate_1 + covariate_2 \dots covariate_n$$
(1)

*Disease_state* corresponds to whether an individual was a case (1) or control (0). The *microbial_feature* variable is the natural-logged relative abundance value for a given species, pathway, or gene family. We logged these values as we found their distributions to be primarily log-normal. *f* represents a fudge factor added to each abundance value for a given microbe, which prevents NAs when computing log values. We chose this value to be the smallest nonzero abundance value for each microbial feature type (species, pathway, and gene family).

In total, two diseases (IBD, otitis) had longitudinal, repeated measured data for certain individuals. To avoid confounding due to intraindividual variation, we averaged the relative abundance of microbial features across all data points for an individual. We additionally averaged any quantitative covariates (e.g., BMI). For categorical covariates (e.g., newborn delivery type, which was encoded as either "cesarean" or "vaginal"), we only kept covariates that were constant across all samples for an individual. We filtered out failed regressions (e.g., singular fits, Supplementary Data 2, column 2) before proceeding with further analyses.

**Meta-analysis**. We performed a meta-analysis over the three diseases for which we had multiple cohorts (CRC, T2D, and adenoma) using the metafor[39] package V2.4.0. We performed a random-effect meta-analysis of the initial regression estimates and standard errors using the "metagen" function (parameters: comb. fixed = FALSE,comb.random = TRUE, method.tau = 'REML',hakn = FALSE, prediction = TRUE, sm = "SMD", control = list(maxiter = 1000)).

We filtered out failed meta-analyses before proceeding with further analyses, and we only meta-analyzed over features that were present in at least two of the cohorts for a given disease. For example, consider T2D, which had two cohorts: if the regression for a given feature failed in one cohort but succeeded in the other, we removed that feature from consideration for T2D, as its failure to reproduce a successful regression, statistically significant or not, we considered a mark against its reproducibility, and therefore utility, as a disease indicator.

**Multiple-hypothesis correction**. We adjusted for false-discovery rate within each disease before selecting statistically significant microbial features to conduct vibration-of-effects analysis on. We combined all feature-fit outputs for a given microbial data type (e.g., species), and computed an adjusted *P* value using the Benjamini–Yekutieli (BY) method and set our significance threshold at an adjusted *p*-value of less than or equal to 0.05. The end result was three different FDR cutoffs, one for each microbial feature type (i.e., species, pathways, and genes).

**Microbial feature selection for vibration of effects**. We selected all FDR-significant features for our vibration-of-effects analysis. We additionally carried out a systematic literature review to identify prior-reported, species-level disease associations for inclusion. Supplementary Data 3 contains a list of species selected for analysis as well as the parameters for our literature review.

**Vibration of effects**. We computed the vibration of effects for a selected microbial feature by fitting a single model for every possible combination of covariates in the maximal model. For example, suppose the maximal model for a given disease and given feature were (using the same variable definitions as in Eq. (1)):

$$ln(microbial\_feature + f) \sim disease\_state + age + sex + BMI$$
(2)

We would then compute $2^n$ models, where $n$ is the number of covariates other than *disease_state*. In this case, $n = 3$, so we would fit a total of eight models, the remaining seven of which are specified below:

$$ln(microbial\_feature + f) \sim disease\_state$$
(3)

$$ln(microbial\_feature + f) \sim disease\_state + age$$
(4)

$$ln(microbial\_feature + f) \sim disease\_state + age + sex$$
(5)

$$ln(microbial\_feature + f) \sim disease\_state + age + sex + BMI$$
(6)

$$ln(microbial\_feature + f) \sim disease\_state + age + BMI$$
(7)

$$ln(microbial\_feature + f) \sim disease\_state + sex$$
(8)

In the case of repeated measures, we executed the same averaging-across-individuals strategy as described above in "*Computing initial microbial feature*

*associations*." For downstream analysis, we extracted *P* values and association sizes (beta-coefficients) for the *disease_state* variable, using the same microbial feature-type FDR threshold as determined in the meta-analyses to gauge statistical significance.

The only case in which we did not fit every possible model was for one of the T2D cohorts, which, given the number of potential adjusters, yielded millions of possible models. Given that we were to vibrate over many thousands of features associated in our initial meta-analysis with T2D, we found computing so many models for each one to be computationally intractable. As such, we selected, for each feature, 50,000 models to fit at random.

**Evaluation of vibration of effects**. We used two metrics to evaluate the vibration of effects when modeling disease–microbiome relationships.

1. Estimate Range (ER)—The difference between the 1st and 99th percentiles of *disease_state*'s beta-coefficient range across all models fit for a given microbial feature association with *disease_state*.
2. Janus Effect (JE)—The fraction of *disease_state* association sizes greater than zero across all model fits.

Feature "robustness" is inversely correlated to estimate range and *P* value range, and increases as the Janus effect approaches 1 or 0 (with maximal Janus effect being at 0.5, where half of the estimates were greater than zero and half were less than zero).

We additionally filtered out pathways labeled as "UNINTEGRATED" or "UNMAPPED" at this point in the pipeline.

**Computing similarity between microbiome-disease architectures**. We constructed a binary matrix, with diseases on the columns and microbial features on the rows, where values of 1 corresponded to a feature being associated with a given disease. To compute similarity in microbiome-disease signatures, we calculated the Jaccard distance between pairwise combinations of columns between species-level features, pathway-level features, gene family-level features, and all three combined. For ease of visualization, we took the natural log of these values when plotting in the heatmap in Fig. 4.

**Mapping genes to taxonomy and construction of phylogenetic tree**. Each of the gene families in our analysis has an associated UniRef90 ID[40], which in turn has an associated UniProtKB ID, which itself has an NCBI taxonomy ID for the taxa in which a given gene family has been observed. We used the R package "taxonomizr" V0.5.0 to map these IDs to NCBI taxonomy strings. We then built a phylogenetic tree based on these taxonomic IDs using the ETE3 Toolkit V3.1.2[41] in Python V3.7[41]

**Comparison of our approach to other methods and permutation tests**. For each phenotype, we compared how the architectures highlighted by our pipeline—fitting maximal models, meta-analyzing where possible, and modeling vibration of effects—would compare, both in statistical significance and strength of association—to 5 alternative approaches: a univariate regression, a linear model only adjusting for cohort batch (where possible), an elastic net, random forest, and sparse partial least squares regression (sPLS) (Supplementary Fig. 3). We fit these models using the caret V6.086 package and the mixOmics V6.12.2 package for sPLS. We did this for all data types (species, pathways, and gene families). We additionally performed permutation tests for every model comparison done, where we randomized the binary disease variable and compared the output (which should be null) to what we found in our architectures. To avoid having to feed millions of features into the elastic net, random forest, and sPLS, for pathways and gene families, we only used the most 10,000 nominally significant associations in the univariate/batch-adjusted regression. For the univariate/batch-adjusted regression, we compared the overlap between significant features to those we found to be robust and significant in our pipeline. For all methods, we compared the concordance (Spearman correlation) between the ranking of features in terms of absolute value of estimate size or, for the random forest, relative importance. The estimate size we used for the features in our architectures was that of either the initial meta-analysis or the initial maximal model, depending on if it were a multicohort phenotype or otherwise. Finally, we additionally compared the feature ranking correlation between only the estimate sizes for the maximal models for the features we included in our architectures and the other methods.

**Validation of gene-level architectures for CRC and IBD in external datasets**. We downloaded additional publicly available IBD (Lloyd-Price et al.)[24] and CRC (Wirbel et al.)[18] cohorts. The latter contained some samples that were already in our analyses. After removing overlapping samples, and ended up with 82 samples in total, 22 cases and 60 controls. The IBD cohort had 106 subjects (26 controls, 80 cases from 1338 subjects). We downloaded the FASTA sequences for the genes that were in our initial CRC and IBD architectures and had representative sequences. We aligned raw reads from our samples back to these genes using Diamond[42] with the default settings. We calculated relative abundance across all genes normalizing by gene length and total reads in a sample, as is the standard in the literature[36]. The IBD cohort was longitudinal, and as in the previous stage of the analysis, we averaged samples from the same individual. For each sample, we took the natural log of gene abundance (adding a small constant of 0.000001 to the normalized values). For each disease fit a univariate, linear model of the form log(*gene_abundance + constant*) ~ *disease_status*, where disease_status was binary, indicating cases or controls. We report nominally significant (p < 0.05) genes in the results as "validated."

**Plotting and figure generation**. We generated all plots with R's ggplot2[43] package V3.3.2, the exception being forest plots, which we made with the meta[44] package V.13.0. We assembled all figures in Adobe Illustrator.

**Other software information**. All statistical analyses were conducted in *R* > V3.6.0. All compute-intensive analyses (e.g., quantification of vibration of effects) were run on Harvard Research Computing's O2 system.

**Reporting summary**. Further information on research design is available in the Nature Research Reporting Summary linked to this article.

## Data availability
All relevant datasets are publicly available. Those used in the initial analysis can be downloaded from the R package associated with curatedMetagenomicData[34]. We used this package's compilation of the data into single dataframes. The dataset information can be found in the "combined_metadata" file available with the package release. Additional datasets used for validation can be downloaded and accessed at ENA accession PRJEB27928 and https://ibdmdb.org/tunnel/public/summary.html, with additional metadata for the former being available at https://github.com/waldronlab/curatedMetagenomicDataCuration/blob/master/inst/curated/WirbelJ_2018/WirbelJ_2018_metadata.tsv. The UniRef90/UniProtKB database, used for identifying gene-level NCBI taxonomic identifiers, can be found at https://www.uniprot.org/.

## Code availability
Code relevant to the aforementioned analyses is present at https://github.com/chiragjp/microbiome_voe. This repository has a DOI registered with Zenodo and the following citation: Braden T Tierney, Yingxuan Tan, Aleksandar D Kostic, Chirag J Patel, Gene-level metagenomic architectures across diseases yield high-resolution microbiome diagnostic indicators, "microbiome_voe", https://doi.org/10.5281/zenodo.4652931, 2021.

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

## Acknowledgements
We thank Harvard Research Computing for providing compute resources for this work. This research was additionally supported by the National Institute of Diabetes and Digestive and Kidney Diseases (T32 DK110919; P30DK036836-30), the National Science Foundation (1636870), the National Institute of Allergy and Infectious Disease (R01AI127250), the American Diabetes Association (ADA) Pathway to Stop Diabetes Initiator Award #1-17-INI-13, and a Smith Family Foundation Award for Excellence in Biomedical Research.

## Author contributions
B.T.T. and C.J.P conceived the project. B.T.T. and J.T. wrote the project code pipeline. A.D.K. and C.J.P. advised on project progress, statistical methodology, and microbiome analyses. B.T.T. wrote the paper with help from C.J.P., A.D.K., and J.T.

## Competing interests
A.D.K. is a cofounder of and scientific advisor to FitBiomics, Inc. At the time of the writing, C.J.P. was an advisor to XY.ai. B.T.T. and Y.T. report no competing interests.

## Additional information

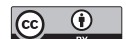

