## [Peer Review File · Nature Communications]

Reviewers' comments:

Reviewer #1 (Remarks to the Author):

The paper "Robust, gene-level, metagenomic architectures across 7 diseases yield high-resolution disease indicators" by Tierney et al. describes a new pipeline to define disease architectures from disease-related metagenome studies using species, pathways and gene abundances as features, per study linear modeling and applying a meta-analysis approach. The authors using a well-curated set of disease-related gut microbiome studies to implement a new framework to extract signatures on different levels that they call 'disease architectures'. They found shared signatures on different description levels between diseases, reproduced known and found new disease associations including strain-specific gene and gene-cluster associations. One major finding is that species-level associations are often unreliable, but gene-level associations are showing distinct signatures but also shared gene clusters between diseases. Species- and pathway-level associations are often dissimilar between disease but gene-level taxonomic descriptions show a broad range of disease and 'health' strain-associations. The authors are showing the advantage of the gene-level analysis for disease-associations of being usable in clinical practice for testing.

In general, the methodology and statistical framework to identify disease associations and extract architectures is new and very interesting. All methods used and described are sound and will be useful for other researchers in the field. The only aspect that is missing in my view is the direct application of the 'architectures' on newer datasets. The underlying curated datasets are from 2018 and since then many studies for the various diseases have been published, e.g. for IBD (Lloyd-Price et al, 2019 Nature). It would be great if the users could show that the architectures hold in those newer data and that they could be used to identify diseases. It would be great to have an application that applies the signatures on any given metagenome. In summary, I think the new framework is an important contribution to the field of host-microbiome associations.

Major points:

- Application on newer datasets to replicate results to show the generalizability of the architectures.
- The limited results for T1D are maybe caused because by one T1D study that the authors used (Heintz-Buschart et al. 2016, Nature Microbiology). This study has a family-based study-design, and the major outcome was that the microbiomes within families are more similar than between families or between T1D patients and healthy individuals. Taking this study as another case-control design seems not appropriate here. Either the authors should take this study out, consider the relationships between individuals for this study or only use unrelated cases and controls.

Minor:

- A permutation test by randomization of disease-label could give additional evidence for the usefulness of the architectures, maybe I missed this in the methods description.
- As the authors also state, their current setting captures only linear relationships. Therefore, it would be interesting to compare the framework to an ML-based method that is able to capture non-linear relationships, but maybe this is beyond the current scope of the paper.
- Heintz-Buschart et al is wrongly referenced in the supplemental tables (as Heitz-BuschartA_2016)

Reviewer #2 (Remarks to the Author):

This paper presents a method for performing meta-analysis of multiple data types with the aim of finding associations between disease states or prognosis and microbiome data. Although the ideas for meta-analysis are creative and likely to bear fruit the current draft lacks appropriate controls, lacks any connection to the large literature in this area, and is far from being acceptable for publication as written.

First and foremost: there is an absolutely enormous literature on meta-analysis in biostatistics. There are far too few connections and comparisons to this prior work. These connections need to be in the form of both citations and explicit comparisons of performance. Compare to sparse-PLS, compare to batch correcting and then running a single model, compare to methods you tell us about that will not work and discuss why in the specific context of this data.

There are no negative controls or benchmarks. The outputs are shown and discussed well, but there are few tests to really show us that the code is working. I appreciate that, with this type of data, there are few benchmarks that are 'plug-and-play' ready to go, but more work on the testing front needs to be done.

You do not discuss compositional effects (CODA, that much of your data is pre-normalized to add up to a pre-set population size) or that there are very strong batch effects that are relevant for covariances as well as absolute abundances.

A strong negative control, like randomizing the disease vector, and other negative controls would help prove the code works.

Reviewer #1 (Remarks to the Author):

The paper "Robust, gene-level, metagenomic architectures across 7 diseases yield high-resolution disease indicators" by Tierney et al. describes a new pipeline to define disease architectures from disease-related metagenome studies using species, pathways and gene abundances as features, per study linear modeling and applying a meta-analysis approach. The authors using a well-curated set of disease-related gut microbiome studies to implement a new framework to extract signatures on different levels that they call 'disease architectures'. They found shared signatures on different description levels between diseases, reproduced known and found new disease associations including strain-specific gene and gene-cluster associations. One major finding is that species-level associations are often unreliable, but gene-level associations are showing distinct signatures but also shared gene clusters between diseases. Species- and pathway-level associations are often dissimilar between disease but gene-level taxonomic descriptions show a broad range of disease and 'health' strain-associations. The authors are showing the advantage of the gene-level analysis for disease-associations of being usable in clinical practice for testing.

In general, the methodology and statistical framework to identify disease associations and extract architectures is new and very interesting. All methods used and described are sound and will be useful for other researchers in the field. The only aspect that is missing in my view is the direct application of the 'architectures' on newer datasets. The underlying curated datasets are from 2018 and since then many studies for the various diseases have been published, e.g. for IBD (Lloyd-Price et al, 2019 Nature). It would be great if the users could show that the architectures hold in those newer data and that they could be used to identify diseases. It would be great to have an application that applies the signatures on any given metagenome. In summary, I think the new framework is an important contribution to the field of host-microbiome associations.

We sincerely thank the Reviewer for their comments and are thrilled they found our methods sound and results important for the field. We are additionally grateful for their critiques, as their integration into our manuscript has, in our opinion, raised its quality and importance even further.

Broadly speaking, in response to Reviewer 1's comments, we 1) removed the T1D dataset from the analysis and 2) additionally carried out validation experiments in additional datasets.

Major points:

1. Application on newer datasets to replicate results to show the generalizability of the architectures.

We scoured the literature for additional datasets in which we could validate specifically our gene-level architectures. Specifically, we were able to find shotgun sequencing data with paired metadata for two phenotypes: colorectal cancer (CRC) and inflammatory

bowel diseases (IBD, using the dataset the reviewer recommended). We chose to move forward with determining how many genes -- of the ~70,000 that were robust and significantly associated between both phenotypes and had representation in UniRef -- were still nominally significant ($p < 0.05$) in a simple univariate, linear association, in these new datasets.

We downloaded the UniRef sequences and their taxonomic annotations, aligning back to them with Diamond and computing their relative abundance on a sample-by-sample basis. We describe this process in the methods:

“Validation of gene-level architectures for CRC and IBD in external datasets

We next aimed to reproduce our gene level architectures for CRC and IBD in independent cohorts. (Wirbel et al. 2019; Lloyd-Price et al. 2019) For IBD, we analyzed 1,338 samples from 106 subjects (80 cases, 26 controls), and for CRC, we analyzed 82 samples from the same number of subjects (22 cases, 80 controls). For the 61,575 genes that had representative sequences in UniRef, were FDR-significant, were robust, and were associated with IBD, 2,104 (3.4%) were nominally significant in the validation cohort. For the 19,498 genes that were FDR-significant, robust, and associated with CRC, 4,688 (24.04%) were nominally significant in the validation cohort (Figure 7A). Of all the associations tested, 10,567 (54.20%) and 22,616 (44.0%) in the validation cohorts had the same direction of association as those in the initial CRC and IBD cohorts. Positive associations with disease were more likely to reproduce than negative in both cases (Figure 7B-C).”

As before, we measured the association between disease status (CRC case vs control, IBD case vs control) and the abundance of each gene. Broadly speaking, we were able to reproduce the taxonomic structures of the architectures. We tentatively validated more genes in CRC than IBD. We additionally found that positive associations with disease (i.e. genes enriched in cases) were easier to reproduce than negative associations.

We report our results in the following Figure (Figure 7), which we reproduce below, with its legend:

Figure 7: Validation of gene-level architectures for CRC and IBD. We tested the association between gene abundance and disease state for the genes associated with IBD and CRC in two cohorts not analyzed in our initial study. A) Overlaps between the initial and validation cohorts in terms of significant genes. B-C) Volcano plots of CRC and IBD estimate sizes and p-values for validation cohorts. Each point represents a different gene. Dotted line is nominal ($p\text{-value} < 0.05$) significance. Black dots indicate the association direction (e.g. positive vs. negative) matched in the initial cohort(s), grey indicates initial and validation associations did not have the same direction. D-E) The top 25 annotations (by frequency) of genes associated with CRC and IBD in the initial cohorts and how many of these are also found in the top 25 annotations of genes in the validation cohorts. A missing blue circle indicates that species was not found in the top 25 annotations for the validation cohort.

We additionally describe these findings in a new Results section:

“Gene-level metagenomic architectures are reproducible across additional cohorts

We next aimed to reproduce our gene level architectures for CRC and IBD using additional publicly available datasets. For IBD, we analyzed 1,338 samples from 106 subjects (80 cases, 26 controls), and for CRC, we analyzed 82 samples from the same number of subjects (22 cases, 80 controls). For the 61,576 genes that were FDR-significant, robust, and associated with IBD, 2,104 (3.4%) were nominally significant in the validation cohorts. For the 20,995 genes that were FDR-significant, robust, and associated with CRC, 4,688 (22.3%) were nominally significant in the validation cohort (Figure 7A). Of all the associations tested, 11,321 (63.02%) and 22,616 (44.0%) in the validation cohorts had the same direction of association as those in the initial CRC and IBD cohorts, respectively, with the majority being underpowered to achieve nominal significance. Positive associations with disease were more likely to reproduce and negative in both cases (Figure 7B-C).

We examined the taxonomic annotations for the genes in addition, counting the number of annotations to each species/strain, and identifying the top 25 for the initial and validation cohorts (Figure 7D-E). We found that the top annotations were similar between the two, with 21 of the top 25 annotations in the validation cohort being present in the initial cohort for CRC. 16 of the top 25 annotations in the validation cohort were present in the initial cohort for IBD. Overall, this indicates that the gene-level architectures, when mapped to strains/species were reproducible in new cohorts.”

And we discuss these results’ interpretations in the Discussion:

“Overall, we found striking and previously unrecognized high-resolution genetic and taxonomic signatures associated with ACVD, IBD, CRC, and cirrhosis, reproducing the taxa that dominated those for CRC and IBD.”

2. The limited results for T1D are maybe caused because by one T1D study that the authors used (Heintz-Buschart et al. 2016, Nature Microbiology). This study has a family-based study-design, and the major outcome was that the microbiomes within families are more similar than between families or between T1D patients and healthy individuals. Taking this study as another case-control design seems not appropriate here. Either the authors should take this study out, consider the relationships between individuals for this study or only use unrelated cases and controls.

We agree with the reviewer that the family-based study design is not a standard case-control study. We also agree it is prudent to remove T1D from our analysis and have done so. We now write in the methods:

“We included the Type 1 Diabetes data from curatedMetagenomicData in our initial regressions, however family-based structure of one cohort (ie one cohort contained data on related individuals) and low sample size of cases prompted removal, specifically to ensure proper harmonization across case-control studies.”

Minor:

3. A permutation test by randomization of disease-label could give additional evidence for the usefulness of the architectures, maybe I missed this in the methods description.

Reviewer 2 also commented on this point, and we have now done this in our model comparison analysis, which is described in detail below.

In short, we carried out an extensive analysis comparing our association method (fitting a maximal model followed by modeling vibration of effects) to a variety of others, including univariate associations, associations adjusted for cohort batch effects, a random forest, and elastic net, and sparse partial least squares regression. For each disease and for each datatype (species, pathways, genes), we computed each of these other methods on both a randomized disease vector and a non-randomized disease vector.

We showed that there was no overlap (in terms of FDR-significant genes) between the randomization output and the genes included in our analysis. We mention this in the Results by writing:

“...We found that architectures built on randomized disease incidence data did not overlap with architectures we identified, yielding no significant associations when adjusted for multiple hypothesis testing ...”

We additionally mention this in the Methods by writing:

“We additionally performed permutation tests for every model comparison done, where we randomized the binary disease variable and compared the output (which should be null) to what we found in our architectures.”

And finally, we display these results in Supplementary Figure 4, which is reproduced below in response to Reviewer 2’s point 7.

4. As the authors also state, their current setting captures only linear relationships. Therefore, it would be interesting to compare the framework to an ML-based method that is able to capture non-linear relationships, but maybe this is beyond the current scope of the paper.

We agree that this is currently outside the scope of the paper, especially as modeling vibration of effects inferential estimates (p-values and standard errors on the

coefficients) from a linear model, which are more difficult to obtain for many machine learning and feature selection techniques used in the microbiome. We have devoted another manuscript to the comparison of similar techniques explicitly (Le Goallec and Tierney et al, Plos Comp Bio 2020).

However, given that Reviewer 2 additionally requested comparisons to other techniques, we chose to compare the approaches with the goal of showing how features would be weighted by other non-linear (specifically, a random forest) techniques. We describe these results in detail in response to Reviewer 2's point 4.

5. Heintz-Buschart et al is wrongly referenced in the supplemental tables (as Heintz-BuschartA_2016)

As mentioned above, we have removed this dataset from the analysis.

Reviewer #2 (Remarks to the Author):

We thank Reviewer 2 for their comments, especially those regarding our methods and their place in the literature and differences to other techniques, as we believe responding to these critiques has strengthened our paper immensely.

1. This paper presents a method for performing meta-analysis of multiple data types with the aim of finding associations between disease states or prognosis and microbiome data.

We thank the Reviewer for this comment, as it demonstrated the need for us to hone the take-home messages of this paper. In the following 3 points, we will clarify these here and show how we have done so in the text as well:

First, we would like to clarify that it was not our aim to propose a novel method for performing meta-analysis, and we apologize if our work came across this way. We simply used meta-analyses as they have been used for many years in the microbiome -- to aggregate summary statistics across datasets. Meta-analysis is but one step in our pipeline, along with modeling vibration of effects. Here is a list of other studies that have used meta-analyses in the microbiome space which are similar to our own meta-analysis step:

Duvallet, Claire, Sean M. Gibbons, Thomas Gurry, Rafael A. Irizarry, and Eric J. Alm. 2017. "Meta-Analysis of Gut Microbiome Studies Identifies Disease-Specific and Shared Responses." *Nature Communications* 8 (1): 1784.

Armour, Courtney R., Stephen Nayfach, Katherine S. Pollard, and Thomas J. Sharpton. 2019. "A Metagenomic Meta-Analysis Reveals Functional Signatures of Health and Disease in the Human Gut Microbiome." *mSystems*

4 (4). <https://doi.org/10.1128/mSystems.00332-18>.

Thomas, Andrew Maltez, Paolo Manghi, Francesco Asnicar, Edoardo Pasolli, Federica Armanini, Moreno Zolfo, Francesco Beghini, et al. 2019. "Metagenomic Analysis of Colorectal Cancer Datasets Identifies Cross-Cohort Microbial Diagnostic Signatures and a Link with Choline Degradation." *Nature Medicine* 25 (4): 667–78.

Wirbel, Jakob, Paul Theodor Pyl, Ece Kartal, Konrad Zych, Alireza Kashani, Alessio Milanese, Jonas S. Fleck, et al. 2019. "Meta-Analysis of Fecal Metagenomes Reveals Global Microbial Signatures That Are Specific for Colorectal Cancer." *Nature Medicine* 25 (4): 679–89.

Here is a list of studies detailing and validating the other critical step in our pipeline, modeling vibration of effects:

Ioannidis, John P. A. 2008. "Why Most Discovered True Associations Are Inflated." *Epidemiology* 19 (5): 640–48.

Klau, Simon, Sabine Hoffmann, Chirag J. Patel, John P. A. Ioannidis, and Anne-Laure Boulesteix. 2020. "Examining the Robustness of Observational Associations to Model, Measurement and Sampling Uncertainty with the Vibration of Effects Framework." *International Journal of Epidemiology*, November. <https://doi.org/10.1093/ije/dyaa164>.

Patel, Chirag J., Belinda Burford, and John P. A. Ioannidis. 2015. "Assessment of Vibration of Effects due to Model Specification Can Demonstrate the Instability of Observational Associations." *Journal of Clinical Epidemiology* 68 (9): 1046–58.

We now cite all of these papers in our manuscript in the Introduction:

"The degree to which variation in model specification (e.g. choosing to adjust for certain confounders and not others) changes the relationship between dependent and independent variables has been described as "Vibration of Effects" (VoE).(Ioannidis 2008; Patel et al. 2015; Klau et al. 2020)"

"Meta-analyses are emerging in the microbiome, and have been used to discover new microbiome-disease associations in colorectal cancer.(Armour et al. 2019; Duvallet et al. 2017; Thomas et al. 2019; Wirbel et al. 2019)"

Second, we would like to make very clear that the goal of our publication is to propose the concept and demonstrate the application of microbiome architectures. We are simply using a pipeline that integrates the existing microbiome literature and observational epidemiology literature (for modeling vibration of effects). As we write in the manuscript,

we chose this pipeline out of a desire to be extremely conservative in the features we selected as “disease-associated.”

Second, after filtering for associations that are robust using meta-analysis and vibration of effects, we sought to derive microbiome architectures, which are the set of metagenomic-human phenotype associations, similar to the emerging concept in human genetics. We address the concern and the previous point, we stating the following in the first paragraph of the Introduction:

“Here, we introduce “microbiome architectures”, which, analogous to human genetic architecture,³ are the characteristics of the microbiome which, as a group, correlate with human phenotype differences. More specifically, we compute architecture by identifying the complete set of associations between the microbiome and a given host disease.”

Third, any method that associates features in observational data can be used to create an architecture. This is why we did not include explicit comparisons to other methods in the initial manuscript. We have actually another manuscript published where the aim was to do just this (Le Goallec and Tierney et al, Plos Comp Bio 2020). We have now added text to the Discussion to clarify this as well as the reasoning behind our choice of pipeline:

“To be clear, we do not claim to identify the “best” method for computing architectures. Rather, we aim to propose architectures as a concept and demonstrate one method for their identification that controls for inconsistency in model output due to model specification. There are many options for computing the association between a disease and microbiome feature, ensuring these associations are robust, and meta-analyzing across datasets.”

This reasoning regarding choice of pipeline is why we mentioned, in the Discussion, that other association methods could yield different results:

“Of course, our approach is not without drawbacks. First, it is extremely conservative: we may be filtering out true positive results, particularly at the species and pathway level. For example, it relies on exclusively linear modeling due to its speed and the ease of performing statistical inference on the associations that are output. Many microbiome studies rely on random forests, which can capture non-linearities but can be difficult to perform inference (e.g., ascertain the error around the association estimate) on or interpret (e.g., propose a direction of association).^{24–26} It is possible that a more sophisticated modeling approach would be able to find more associations. We did demonstrate that alternative modeling approaches -- like the elastic net, random forest, and sparse PLS, tend to yield similar results for the features our approach identified, however it was also clear that they would potentially be more sensitive. Moreover, we made choices in our modeling strategy -- like averaging longitudinal samples from the

same individual – due to issues of scaling mixed effect models to the needs of our study -- that could affect statistical power.

For these reasons, we do not claim that our method is the only one to identify architectures -- rather, the value of our work lies in demonstrating the utility of (specifically, gene-level) architectures for comparing and reproducing disease signatures across the microbiome. As with any statistical challenge facing a plethora of analytic choices, the “best” method for identifying those genes remains to be determined and must be the focus of future work.”

2. Although the ideas for meta-analysis are creative and likely to bear fruit the current draft lacks appropriate controls, lacks any connection to the large literature in this area, and is far from being acceptable for publication as written.

We thank the Reviewer for commenting that our approaches are creative and likely to bear fruit, and we have endeavoured to update our manuscript according to their critiques.

3. First and foremost: there is an absolutely enormous literature on meta-analysis in biostatistics. There are far too few connections and comparisons to this prior work.

We agree that the meta-analyses are a widely used approach to synthesize findings across studies. Modern examples, as the reviewer will know, include meta-analyses to summarize therapeutic effects across randomized clinical trials (e.g., (Lau et al. 1992)) and reviewed here (Thacker 1988). Again, we are not claiming to re-invent the process of meta-analyses in observational microbiome studies (essentially a way to estimate an overall estimate across multiple studies (by weighting by the errors across the studies)), but as an off-the-shelf approach to combine associations, and not different from initial meta-analyses that emerged from the microbiome literature (see point 1 for references).

We now mention additional meta-analysis literature, both in the microbiome and beyond, in the Introduction:

“Many studies use “meta-analyses” to aggregate and compare results across cohorts. There are a few approaches for carrying out a meta-analysis (e.g., random versus fixed effects meta-analyses(Borenstein et al. 2009)), and they provide a way to estimate an “overall” association size across cohorts. Historically they have been deployed for both randomized and observational research(Thacker 1988), such as to aggregate effects across clinical trials.(Lau et al. 1992) Meta-analyses are emerging in the microbiome, and have been used to discover new microbiome-disease associations in colorectal cancer. (Armour et al. 2019; Duvallet et al. 2017; Thomas et al. 2019; Wirbel et al. 2019)”

4. These connections need to be in the form of both citations and explicit comparisons of performance. Compare to sparse-PLS, compare to batch correcting and then running a single model, compare to methods you tell us about that will not work and discuss why in the specific context of this data.

In short, we have now carried out an extensive comparison to other modeling approaches.

To clarify, however, we did not mean to imply that we were proposing the “best” or “correct” method for identifying microbiome architectures. As we state in our response to Reviewer 2, point 1, any association method -- from a wilcoxon test to complex machine learning -- can be used to associate a microbiome feature to a phenotype, and an architecture be built from those relationships. We chose our approach due to its 1) conservatism and 2) the ease at which we could extract infer (e.g., attain standard errors and p-values) from data, which is more difficult for other variable selection methods. We at no point state that other potential methods will “not work.” We mention this in the Discussion at:

“For example, it [our approach] relies on exclusively linear modeling due to its speed and the ease of performing statistical inference on its output. Many microbiome studies rely on random forests, which can capture non-linearities but can be difficult to perform inference on or interpret.^{24–26} It is possible that a more sophisticated modeling approach would be able to find more associations.

We generally agree with Reviewer 1 that the comparison to other techniques is outside the scope of this paper, and indeed we have another manuscript devoted to this purpose explicitly (Le Goallec and Tierney et al, Plos Comp Bio 2020). However, since both Reviewers touched on this point, it is clear to use that a comparison to other methods would help readers contextualize our results. As such, we carried out an analysis to demonstrate how architectures may change as a function of modeling strategy.

We describe this analysis (which includes a random forest to touch on Reviewer 1’s point regarding non-linear techniques as well as sparse partial least squares [sPLS] to touch on Reviewer 2’s request) in Supplementary Figure 3, which we reproduce below with its legend:

Supplementary Figure 3: Benchmarking strategy. We compared our modeling strategy (consisting of association, meta-analysis, and vibration of effects) to a number of other potential methods that are frequently used in microbiome studies. Specifically, we aimed to identify if 1) the variable importance/estimate sizes were similar between different methods and 2) if a randomization test (randomizing the disease vector) showed that our or similar approaches would yield large numbers of false positive results. As such, for each data type (top row), and with the disease vector randomized/non-randomized, we computed association for each feature using a univariate linear regression, or, where possible, linear regression adjusted for cohort. To avoid failure due to computational complexity, we then selected the top 10K most significant features and ran the methods in the bottom row, comparing the output to all other steps.

We fit approximately 25 million models in this process and compared the output at all steps. For the variable selection techniques (random forest, elastic net, sPLS), we only looked at the 10,000 most significant features (relevant only for gene families and pathways, which had greater than 10,000 features) in the univariate associations due to the difficulty of fitting such complicated models on millions of features. For these 10,000 features, we compared their feature rankings to the univariate, batch corrected, and VoE-robust and significant beta-coefficient absolute values.

Note that in the above Figure, we also describe the experiment we did randomizing the disease vector as a negative control to see how architectures change. We will fully address those results in Reviewer 2's points 5) and 7).

Overall, we found that fitting a maximal model followed by modeling vibration of effects was, as we expected, more conservative in terms of the number of features that were selected in the architectures. A number of features that would have been statistically significant in a univariate or batch corrected model were filtered out due to being non-robust.

We now include the following Supplementary Figure 5 and legend in the manuscript:

Supplementary Figure 5: Comparing modeling strategies. For the modeling strategies described in Supplementary Figure 3, we identified p-values where possible and compared overlaps in statistical significance between our initial output and other linear modeling methods. We took robust (by modeling VoE) and FDR significant features for the initial maximal model associations in our pipeline and compared those to features that had a nominal p-value less than the p-value equivalent to FDR = 0.05 in our initial approach.

And we reference it in the results, writing:

“We found that architectures (a) built on randomized disease indicators (for cases and controls) did not overlap with architectures we identified, and the (b) other approaches yielded similar importance in variable ranking across all datatypes and phenotypes. This

indicated that generally the same features are being implicated in diverse modeling approaches and that modeling VoE is the most conservative option”

We then measured the concordance (Spearman correlation) between the ranking of features for each method, including the variable selection techniques, which we show in the following figure (Supp Fig 6), reproduced below with its legend:

Supplementary Figure 6: Concordance between regression methods. We measured the estimated concordance in feature ranking (e.g. relative importance or absolute value of beta-coefficient) for each method we compared for (top row) all features used in the comparison analysis for each datatype and phenotype and (bottom row) only the features deemed statistically significant and robust by our initial pipeline. Columns correspond to if the linear model used to select the 10K features for variable selection analysis was adjusted for cohort batch or not. The black box indicates the correlation between the ranking of the absolute value of beta-coefficients from our pipeline for robust features (derived from the maximal models) and the other methods.

We found that all the techniques performed similarly in terms of ranking features, especially when considering only the features included in our final analysis post modeling VoE (second row).

We describe this in the results:

“For the comparison to variable selection methods, univariate models, followed by sPLS and elastic nets, were the most similar to our method. Overall, we found that the ranking of features -- especially among those flagged as significant and robust by our pipeline (mean Spearman correlation with our initial ranking of features = 0.88 +/- 0.01) -- was similar across all methods (Supp Fig 6).”

In summary, we aim for this analysis to show that 1) our approach performs similarly to others “on the market” and that 2) its major difference is that it is simply more conservative due to the fact that we filter out non-robust results.

5. There are no negative controls or benchmarks. The outputs are shown and discussed well, but there are few tests to really show us that the code is working. I appreciate that, with this type of data, there are few benchmarks that are 'plug-and-play' ready to go, but more work on the testing front needs to be done.

We address negative controls below, in Reviewer 2 point 7.

6. You do not discuss compositional effects (CODA, that much of your data is pre-normalized to add up to a pre-set population size) or that there are very strong batch effects that are relevant for covariances as well as absolute abundances.

We thank the reviewer for this comment as we agree that in both our manuscript, as well as in the broader microbiome field, the compositional effects of using relative abundance data to compute associations is often overlooked. Currently, the field tends to use relative abundances instead of absolute abundances. As a result, if microbial genes/species/taxa are not sampled appropriately (e.g. some organisms are oversampled and some are undersampled), their relative abundance will be biased (e.g. the true abundance of microbe “X” in a population might be lesser or greater than represented by its relative abundance). While absolute abundances would potentially address this issue if we could perfectly ascertain true population size, shotgun sequencing is currently unable to achieve this. For example, depth of sequencing, gene length, pathway size, and/or microbial genome size (i.e. number of base pairs) can bias absolute abundances (and in some cases, relative abundances as well). Microbiome researchers are forced to generally assume even sampling of a microbial population and adjust for sequencing depth, calibrating to a pre-set population size.

We agree with the reviewer that the microbiome field needs to consider these points to move forward, and indeed, as sequencing technology the capability of absolute quantification needs to be tested followed by calibration to relative abundance. While we believe that these tests are outside the scope of this paper, we now reference literature (Morton et al. 2019; Barlow et al. 2020) addressing the issues of compositional data in the microbiome and new technologies and standards in the space in the Discussion, writing:

We now address these points in the Discussion, writing:

“Finally, an additional drawback of the data underlying our findings is its reliance on compositional data (i.e. relative abundances). Many have raised issues regarding potential bias of batch effects of relative abundance data (which is a confounder like other variables assessed in our VoE approach). As our ability to ascertain true microbial population size improves via sequencing technology and statistical methods, we should consider leveraging alternative tools or data structures in building architectures to reduce this source of bias. (Morton et al. 2019; Barlow et al. 2020)”

7. A strong negative control, like randomizing the disease vector, and other negative controls would help prove the code works.

We thank both reviewers (Reviewer 1 point 3 and Reviewer 2 points 5 and 7) for addressing the need for negative controls, specifically through randomizing the disease cases and controls. To address this point, for every strategy we tested in our modeling comparison, we additionally carried out a permutation test where we randomized the disease case-control status and re-ran different modeling association strategies. We showed that when using the nominal p-value equivalent to the FDR adjusted p-value done in the VoE analysis, there was little to no overlap in significant features between the randomized univariate and batch corrected regressions with the VoE-robust, meta-analyzed, maximal model output. Indeed, at such a low p-value threshold, there were no significant features at all for the randomized data analysis. As a result, we conclude that our architectures are providing distinct, and biologically meaningful, results when compared to randomized disease vectors.

We reference these results in the manuscript Results section:

“We additionally randomized disease status for all diseases as a negative control. We found that architectures (a) built on randomized disease indicators (for cases and controls) data did not overlap with architectures we identified (Supp Fig 4)”

And we additionally include the following new supplementary figure with its legend:

Supplementary Figure 4: Permutation experiment. As a negative control, for the modeling strategies described in Supplementary Figure 3, we randomized the disease variable and identified p-values where possible and compared overlaps in statistical significance between our initial output and other linear modeling methods. We took robust (by modeling VoE) and FDR significant features for the initial maximal model associations in our pipeline and compared those to features that had a nominal p-value less than the p-value equivalent to FDR = 0.05 in our initial approach.

We additionally would like to comment on another form of a biological “negative” control, the inclusion of otitis in our analysis. Otitis has not, to our knowledge, been linked to the gut microbiome before. Given its pathology (affecting the ear), we would not hypothesize it having a strong gut microbial signature compared to other phenotypes, like IBD. We found this to be the case, and now in the Introduction mention this point:

“We specifically chose to examine otitis as a form of negative biological control, as, to our knowledge, it has limited reported association with the gut microbiome, and we expected it to have a negligible architecture.”

We additionally mention this in the Discussion:

“We additionally found limited associations with otitis, which we hypothesized would be the case due to its limited reported biological link to the gut microbiome.”

Finally, we would like to address Reviewer 2’s concern that our code works. We included our code in a GitHub repository upon initial submission, replete with a usable example for Reviewer’s and readers to access:

https://github.com/chiragjp/microbiome_voe/tree/master/example_scripts

Note, however, to avoid confusion that we were releasing a package, we write at the top of the README:

“This code is NOT meant to be distributed or deployed as a package, and instead is meant to provide assistance in understanding and reproducing the specific work in our manuscript. If you’re interested in running VoE analyses yourself on other observational data, we are currently developing an R package for just this task that can be accessed at <https://github.com/chiragjp/quantvoe>”

Of course, just because we are not “packaging” this code, we of course needed to validate that it works. This is why we included our example with CRC as well as detailed instructions as to how to run it in the README. In this example, we:

- 1) Compute maximal model associations between disease and microbial features
- 2) Meta-analyze across the features
- 3) Run vibration of effects (fitting all possible models)
- 4) Save and plot the output

Each of these steps provides the output of each model fit as well as the initial data. The exact raw data, variables, and model specification for every model can be accessed in the output data. If there were concerns regarding the accuracy of our code, outside of going through the functions themselves, the exact model run for each vibration could be easily reproduced.

Given the 1) ease of querying our output in the example dataset, 2) example code, 3) the concordance between our results and other modeling strategies, 4) our ability to reproduce the biological literature (e.g. associations between CRC and *F. nucleatum*), 5) the lack of associations with otitis, and, 6) the success of our permutation test for the disease variable described above, we feel that our code is well up to the publishable standard.

REVIEWERS' COMMENTS

Reviewer #1 (Remarks to the Author):

The authors have addressed all my comments and suggestions in the new version of the manuscript.

Reviewer #2 (Remarks to the Author):

The authors have done a tremendous amount of additional work. Although i was quite harsh in my first review I feel that the authors have addressed most of my concerns. A few substantive differences of opinion remain, but none that are critical.